



# Variations of China's emission estimates response to uncertainties in energy statistics

Chaopeng Hong[1,2], Qiang Zhang[1,5], Kebin He[2,4,5], Dabo Guan[1,3], Meng Li[1,2], Fei Liu[1,2], and Bo Zheng[2]

[1]Ministry of Education Key Laboratory for Earth System Modeling, Center for Earth System Science, Tsinghua University, Beijing, China
[2]State Key Joint Laboratory of Environment Simulation and Pollution Control, School of Environment, Tsinghua University, Beijing, China
[3]School of International Development, University of East Anglia, Norwich NR4 7TJ, United Kingdom
[4]State Environmental Protection Key Laboratory of Sources and Control of Air Pollution Complex, Beijing, China
[5]Collaborative Innovation Center for Regional Environmental Quality, Beijing, China

*Correspondence to*: Qiang Zhang (qiangzhang@tsinghua.edu.cn)

**Abstract.** The accuracy of China's energy statistics is of great concern because it contributes greatly to the uncertainties in estimates of global emissions. This study attempts to improve the understanding of uncertainties in China's energy statistics and evaluate their impacts on China's emissions during the period of 1990-2013. We employed the Multi-resolution Emission Inventory for China (MEIC) model to calculate China's emissions based on different official datasets of energy statistics using the same emission factors. We found that the apparent uncertainties (maximum discrepancy) in China's energy consumption increased from 2004 to 2012, reaching a maximum of 646 Mtce (million tons of coal equivalent) in 2011, and that coal dominated these uncertainties. The discrepancies between the national and provincial energy statistics were reduced after the three economic censuses conducted during this period, and converging uncertainties were found in 2013. The emissions calculated from the provincial energy statistics are generally higher than those calculated from the national energy statistics, and the apparent uncertainty ratio (the ratio of the maximum discrepancy to the mean value) owing to energy uncertainties in 2012 took values of 30.0%, 16.4%, 7.7%, 9.2% and 15.6%, for $SO_2$, $NO_x$, VOC, $PM_{2.5}$ and $CO_2$ emissions, respectively. $SO_2$ emissions are most sensitive to energy uncertainties because of the high contributions from industrial coal combustion. The calculated emission trends are also greatly affected by energy uncertainties - from 1996 to 2012, $CO_2$ and $NO_x$ emissions, respectively, increased by 191% and 197% according to the provincial energy statistics but by only 145% and 139% as determined from the original national energy statistics. The energy-induced emission uncertainties for some species such as $SO_2$ and $NO_x$ are comparable to total uncertainties of emissions as estimated by previous studies, indicating variations at energy consumption could be an important source of China's emission uncertainties.



## 1 Introduction

China is facing a considerable challenge related to cleaning its air (Zhang et al., 2012). Emission inventories of air pollutants and greenhouse gases are of fundamental importance for the scientific analysis of complex air pollution problems and climate change as well as for assisting policy-makers in designing mitigation policies. Reliable emission inventories are becoming increasingly important, especially for large and rapidly growing countries such as China. To date, emissions have generally been estimated based on bottom-up approaches that combine available statistical information on relevant activities with known emission factors for different sectors and fuel types. Although a number of emission inventories covering China have been conducted, such as TRACE-P (Streets et al., 2003), INTEX-B (Zhang et al., 2009), MEIC (http://www.meicmodel.org/), REAS (Ohara et al., 2007; Kurokawa et al., 2013), EDGAR (http://edgar.jrc.ec.europa.eu/index.php) and GAINS (http://gains.iiasa.ac.at/models/), China's emission inventories are thought to be quite uncertain because of uncertainties in activity-related data, such as energy consumption data, and a lack of local emission factors (Zhao et al., 2011).

China has now become the world's top consumer of primary energy; however, the reliability of China's energy statistics has frequently been questioned (Sinton, 2001; Akimoto et al., 2006; Guan et al., 2012). The accuracy of China's energy statistics is of great concern because it contributes greatly to uncertainties in estimates of global emissions (Marland et al., 2012). Several inconsistencies exist among different sets of official energy statistics, namely, the national (CT-CESY, country-total) and provincial (PBP-CESY, province-by-province) Energy Balance Sheets from the China Energy Statistical Yearbook (CESY) and the Energy Balance Sheets from the International Energy Agency (IEA). These inconsistencies in energy consumption may lead to significant discrepancies in China's emission estimates. As previously reported (Akimoto et al., 2006), the increases in $NO_x$ emissions estimated based on the PBP-CESY and IEA2004 data from the 1996-2002 period are 25% and 15%, respectively, and that estimated from the CT-CESY data is even lower. Zhao et al. (2011) used Monte Carlo methods to quantify the uncertainties of a bottom-up inventory of Chinese anthropogenic atmospheric pollutants and found that emission factors, rather than activity levels (e.g., energy consumption), are the main source of uncertainties in Chinese emission estimates. However, relatively small uncertainties in the activity levels for the year 2005 (i.e., CVs of 5%, 10% and 20% for the activity levels of the power sector, industrial combustion and residential fossil fuel use) were considered in their study. Some studies have noted the large uncertainties in energy statistics in recent years and their impacts on $CO_2$ emission estimates (Guan et al., 2012; Liu et al., 2015; Korsbakken et al., 2016). Guan et al. (2012) found that $CO_2$ emissions calculated on the basis of two publicly available energy datasets (i.e., CT-CESY and PBP-CESY) for 2010 differ by 1.4 gigatons, which is equivalent to approximately 5% of the global total. Liu et al. (2015) estimated that total energy consumption in China was 10 per cent higher in 2000–2012 than the value reported by China's national statistics. Korsbakken et al. (2016) used correlated economic quantities to constrain growth rates in total coal-derived energy use. They pointed out uncertainties around reductions in China's coal use and $CO_2$ emissions in recent years, and questioned the 2.9% drop in Chinese coal consumption in 2014 in preliminary official statistics, and showed that it was inappropriate for



estimating $CO_2$ emissions. Previous studies on the uncertainties in China's energy statistics and emissions are typically applicable either to an early period or for only a few species (usually $CO_2$ and $NO_x$).

This paper strives to present a full evaluation of the uncertainties in China's energy statistics and their effects on emission estimates for China during the period from 1990 to 2013. The evaluated species include $SO_2$, $NO_x$, VOC, $PM_{2.5}$ and $CO_2$. In this study, apparent uncertainties in China's energy statistics were evaluated through detailed comparisons of publicly available energy statistics to provide indirect but still useful information regarding the range of uncertainty of existing energy activity data. We defined the apparent uncertainty as the maximum discrepancy among different datasets and the apparent uncertainty ratio as the ratio of the maximum discrepancy to the mean value from the different datasets. To evaluate the impact of these energy uncertainties on China's emissions and the emission trends, we established several emission inventories based on these energy statistics in the framework of the MEIC inventory using the same emission factors.

This paper is organized as follows. Section 2 summarizes the methods and data that were used in this work, including the energy statistics for China, and the MEIC emission inventory. In Sect. 3, we evaluate the apparent uncertainties in China's energy statistics and their impacts on China's emissions and the emission trends. In Sect. 4, we discuss the reliability of China's energy statistics and the implications for other inventories.

## 2 Data and methods

### 2.1 China's energy statistics

China publishes its official energy statistics annually in the China Energy Statistical Yearbook (CESY) released by the National Bureau of Statistics (NBS), including both national and provincial Energy Balance Sheets for each province. The national Energy Balance Sheets are revised each time an economic census is completed, and the revisions are published in the next Energy Statistical Yearbook. The China Energy Statistical Yearbooks from 2005 (CESY2005), 2009 (CESY2009) and 2014 (CESY2014) contain the revised national energy data for the periods of 1999-2003, 1996-2007 and 2000-2012, respectively, based on the results of the first, second and third national economic censuses conducted during this time. The International Energy Agency (IEA) also publishes energy statistics for China, which have been widely used in international emission inventories (such as EDGAR). The IEA also regularly revises its energy statistics and is now operating in cooperation with the NBS, who annually provides the IEA with China's energy statistics, and in recent years, the IEA statistics are found to be quite consistent with the NBS's national Energy Balance Sheet.

In this study, we used six datasets of energy statistics: the original edition of the national Energy Balance Sheets from the CESY (CT-CESY-Ori) and its revisions following the first economic census (CT-CESY-1C), the second economic census (CT-CESY-2C) and the third economic census (CT-CESY-3C); the provincial Energy Balance Sheets from the CESY (PBP-CESY); and the 2012 edition of China's energy statistics from the IEA (CT-IEA-2012). These datasets are summarized in Table 1. Note that here CT-CESY-Ori represents the first edition of national energy statistics covering the whole period





1990-2013. For revised national energy statistics (i.e., CT-CESY-1C, CT-CESY-2C, CT-CESY-3C), the data were taken from previous edition for years that revised data were unavailable. Although energy statistics for 2014 is already published, we did not include year 2014, for the reason that the emission inventory is being updated.

## 2.2 Emission inventory

The MEIC emission inventory model (available at http://www.meicmodel.org) was used in this study to investigate the emission responses to different energy statistics. MEIC is a dynamic technology-based inventory developed for China covering the years from 1990 to 2013 by Tsinghua University following the work of INTEX-B (Zhang et al., 2009), with several updates, such as a unit-based emission inventory of power plants (Liu et al., 2015), a high-resolution vehicle emission inventory at the county level (Zheng et al., 2014), and an improved NMVOC speciation approach for various chemical mechanisms (Li et al., 2014). MEIC inventory includes recent control policies based on the available official reports (Ministry of Environmental Protection of China (MEP), 1991-2014, 2000-2014). The MEIC version 1.1 (MEIC v1.1) uses energy consumption data from PBP-CESY, excluding diesel and gasoline consumption data, which are taken from the national energy statistics (currently CT-CESY-1C) because the diesel consumption data provided in the national energy statistics were thought to might be more reliable (Zhang et al., 2007). To further explore the impact of energy data inconsistencies on estimates of China's emissions, five emission inventories based on five sets of energy statistics (i.e., CT-CESY-Ori, CT-CESY-1C, CT-CESY-2C, CT-CESY-3C and PBP-CESY) were established in the framework of the MEIC inventory. Note that only energy data were changed in the calculations of these emission inventories, while other data such as emission factors remained the same as MEIC inventory. Thus the emission uncertainties derived from these inventories are only those associated with energy uncertainties. They do not include uncertainties in the emission factors and other parameters in MEIC inventory. The IEA energy statistics were excluded from the emission calculations because they are based on NBS's national Energy Balance Sheets, and currently quite consistent with CT-CESY-2C. They may soon be updated based on CT-CESY-3C.

## 3 Results

### 3.1 Apparent uncertainties in China's energy statistics

Apparent uncertainties in China's energy consumption for the period of 1990-2013 were quantified based on publicly available energy statistics (i.e., CT-CESY-Ori, CT-CESY-1C, CT-CESY-2C, CT-CESY-3C, PBP-CESY and IEA2012), as shown in Fig. 1. Before 1996 there are no annual provincial data, and essentially just one national data set, which has not been revised. But since 1996, multiple data sets and/or revisions are available for each year. Notable apparent uncertainties have been observed since then, which can be divided into three periods: an early period (1996-2003); a more recent period of rapid growth (2004-2012); and the most recent period of convergence (2013). During the early period (1996-2003), China's energy consumption grew slowly, from 1352-1389 Mtce in 1996 to 1709-1971 Mtce in 2003. The average apparent



uncertainty in total energy consumption during this period is 133 Mtce, with a peak of 261 Mtce in 2003, and the corresponding apparent uncertainty ratios are 9.0% for the period as a whole and 14.3% for 2003. During the recent period of rapid growth (2004-2012), along with the rapid growth in China's economy and energy consumption, the apparent uncertainty in the total energy consumption also increased, with a mean uncertainty of 453 Mtce for this period and a

maximum of 646 Mtce in 2011; the corresponding apparent uncertainty ratios are 14.7% for this period overall and 16.9% for 2011. The inconsistencies during the early period have been reported in many previous studies (Sinton, 2001; Akimoto et al., 2006; Zhang et al., 2007), but few studies (Guan et al., 2012; Liu et al., 2015; Korsbakken et al., 2016) have noted the more recent rapid growth period. Converging uncertainties are observed in 2013, with the release of the newest energy statistics based on the third economic census—the apparent uncertainty in total energy consumption for 2013 is reduced to

62 Mtce, and the corresponding apparent uncertainty ratio is only 1.5%. We notice that the apparent uncertainty for 2014 (not shown here) is similar to that for 2013, also much smaller than that during the recent period of rapid growth (2004-2012).

With regard to different types of energy, coal dominates the apparent uncertainties in total energy consumption. The average apparent uncertainties in coal consumption for 1996-2003, 2004-2012 and 2013 are 147 Mtce, 429 Mtce and 194 Mtce,

respectively, and the corresponding apparent uncertainty ratios are 14.2%, 19.8% and 6.7%. The sum of the provincial data (PBP-CESY) is generally higher than the national total (i.e., CT-CESY-Ori, CT-CESY-1C, CT-CESY-2C and CT-CESY-3C) with regard to total energy consumption and coal consumption. After each of the three economic censuses, the national total energy consumption data (CT-CESY-1C, CT-CESY-2C and CT-CESY-3C) were revised upward to approach the provincial totals, primarily by adjusting the coal-related data. The recent edition of IEA energy statistics (IEA2012) is found to be

similar to CT-CESY-2C. The apparent uncertainties in oil consumption during 1996-2003 are relatively large, with a mean of 48 Mtce and an average apparent uncertainty ratio of 15.7%. The provincial total oil consumption is lower than the national total for 1996-2003, but this situation is reversed between 2005 and 2011. The apparent uncertainties in the consumption of natural gas and other types of energy are smaller than the uncertainties in coal and oil, suggesting that the statistical data for natural gas and other energy sources may be more accurate because their use is generally metered.

The apparent uncertainties in coal consumption were further analyzed by sector, as shown in Fig. 2. As the largest consumer of coal in China, the power sector is found to exhibit less uncertainty in its coal consumption than other sectors. Coal consumption in the industrial sector is highly uncertain, with an apparent uncertainty ratio for 2012 of 45.4%, which represents the greatest contribution to the total uncertainty in coal consumption. A significant decrease in coal consumption in the industrial sector during 1996-2002 is observed in the CT-CESY-Ori data, and this decrease resulted in a slight

decrease in the total coal consumption. For the heating sector and the residential sector, although the levels of coal consumption in these two sectors are smaller than those in the power and industrial sectors, comparable apparent uncertainties are also found; the apparent uncertainty ratios for the heating and residential sectors in 2012 are 37.8% and 46.9%, respectively.





### 3.2 Effects on China's emission estimates

To evaluate the effects of uncertainties in the energy statistics on China's emission estimates, five emission inventories based on different sets of energy statistics (i.e., CT-CESY-Ori, CT-CESY-1C, CT-CESY-2C, CT-CESY-3C and PBP-CESY) were established. Figure 3 shows the apparent uncertainties in China's emissions during 1990-2013. Figure 4 shows the apparent uncertainty ratio in China's emissions during 1990-2013. It should be noted that the emission uncertainties discussed below, which were derived from these five emission inventories, are based only on uncertainties in the energy data; thus, they could reflect the impacts of energy uncertainties on emission estimates. For the early period (1996-2003), the average apparent uncertainties for $SO_2$, $NO_x$, VOC, $PM_{2.5}$ and $CO_2$ are 2.10 Tg, 0.83 Tg, 0.41 Tg, 0.34 Tg and 278 Tg, respectively, and the corresponding apparent uncertainty ratios are 10.2%, 6.7%, 3.2%, 2.8% and 6.7%. For the recent period of rapid growth (2004-2012), the apparent uncertainties are increasing over time and are more significant than those in the early period, although this fact has rarely been discussed in the literature; the average apparent uncertainties during this period for $SO_2$, $NO_x$, VOC, $PM_{2.5}$ and $CO_2$ are 5.77 Tg, 2.98 Tg, 1.60 Tg, 0.80 Tg and 1026 Tg, respectively, and the corresponding apparent uncertainty ratios are 20.4%, 12.6%, 7.7%, 6.4% and 12.4%. For 2012, the apparent uncertainties for these species are 7.76 Tg, 4.68 Tg, 1.90 Tg, 1.10 Tg and 1633 Tg, respectively, and the corresponding apparent uncertainty ratios are 30.0%, 16.4%, 7.7%, 9.2% and 15.6%. The apparent uncertainty for $CO_2$ in 2010 is 1283 Tg in this study, which is similar with the discrepancy (~1400 Tg) reported by Guan et al. (2012), but lower than the uncertainty in 2012. In the most recent period of convergence (2013), the apparent uncertainty ratio in emissions is less than 5% for most species because of the lower uncertainties in the energy statistics after the third economic census. Note that the emission discrepancies calculated from the provincial and national energy statistics are getting smaller after the third economic census (i.e., PBP-CESY and CT-CESY-3C), compared with that before the third economic census (e.g., PBP-CESY and CT-CESY-2C). For example, $CO_2$ emission discrepancy in 2010 between PBP-CESY and CT-CESY-3C is only 548 Tg, much less than that reported by Guan et al. (2012), in which the NBS's data before the third economic census were used.

As seen, the energy-induced uncertainties in emissions differ by species; the largest uncertainties are observed for $SO_2$, followed by $NO_x$ and $CO_2$, and the smallest are found for $PM_{2.5}$ and VOC. Taking the year 2012 as a case in which the uncertainties are prominent, the emission uncertainties were separated by sector and by energy type, as shown in Table 2. $SO_2$ emissions are more sensitive to energy uncertainties than are $CO_2$ emissions because of the high contribution (approximately 50%) from industrial coal combustion, which is the largest source of uncertainty in $SO_2$ emissions (6.04 Tg). A large fraction (approximately 24%) of $NO_x$ emissions is contributed by the use of diesel in the transportation sector; the corresponding activity data have a lower uncertainty ratio than that for coal use, leading to a lower sensitivity than that of $SO_2$. $PM_{2.5}$ and VOC emissions also show less sensitivity to energy uncertainties because they represent relatively small contributions from energy consumption and high contributions (approximately 70%) from industrial process emissions. With regard to contributions by sector, industry is the dominant sector, accounting for 77.8%, 72.3%, 46.8%, 52.4% and 73.2% of the total apparent uncertainties in $SO_2$, $NO_x$, $PM_{2.5}$, VOC and $CO_2$ emissions, respectively. Although the power sector is a



major source of emissions of many species (contributing approximately 25-35% of the total emissions of $CO_2$, $NO_x$ and $SO_2$), it is estimated to contribute less than 7% to the total apparent uncertainties for all species because of the relatively low uncertainty for coal consumption in the power sector. Transportation is another key contributor to emission uncertainties for $NO_x$ (contributing 17.3% of the uncertainty for this species), whereas the residential sector is significant for $SO_2$

(contributing 18.3% of the uncertainty). With regard to energy type, 97.6% and 93.8% of the emission uncertainties of $SO_2$ and $PM_{2.5}$, respectively, originate from coal, whereas 31.2% of the VOC emission uncertainties come from oil. The contributions of gas and other fuels are negligible because their emissions are relatively small.

Discrepancies in energy data affect not only the absolute emission estimates for individual years but also multi-year emission trends because of the inter-annual variability of these discrepancies. Table 3 compares the emission trends for China derived

from different energy statistics. For the early period (1996-2003), slower growth rates of $CO_2$, $NO_x$ and $SO_2$ emissions are found from the CT-CESY-Ori inventory (22.9%, 38.0% and 14.3%, respectively) than from the PBP-CESY inventory (35.5%, 47.5% and 28.0%, respectively), which is consistent with previous studies (Akimoto et al., 2006; Zhang et al., 2007). The trends derived from the national energy statistics were revised upward after each of the three economy censuses, bringing them closer to those indicated by the provincial energy statistics. $SO_2$ and $CO_2$ show a dip according to the CT-

CESY-Ori inventory, but this effect is not significant in the PBP-CESY inventory. $SO_2$ emissions declined by 13.7% during the period of 1996-2000 according to the CT-CESY-Ori inventory but increased by 1.4% according to the PBP-CESY inventory. These differences reflect the large uncertainties in industrial coal consumption during 1996-2000—a decline of 28.4% is indicated by CT-CESY-Ori, whereas only a slight decrease of 3.8% is found from PBP-CESY. It should be noted that the dip for $NO_x$ emissions is not as distinct as that for $CO_2$. This is because the fuel consumption in the power and

transportation sectors, for which the $NO_x$ emission factors are the largest, was steadily increasing during this period.

In the recent period of rapid growth (2004-2012), $CO_2$ and $NO_x$ emissions, respectively, increased by 91.8% and 77.6% according to the PBP-CESY inventory but by only 70.8% and 48.4% as determined from the CT-CESY-Ori inventory; $SO_2$ emissions increased by 1.6% according to the PBP-CESY inventory but decreased by -18.8% as indicated by the CT-CESY-Ori inventory. For the period from 1996 to 2012, the $CO_2$ growth rates inferred from the CT-CESY-Ori, CT-CESY-3C and

PBP-CESY inventories are 145%, 172% and 191%, respectively, similar with the growth rates in the total energy consumption (160%, 197% and 207%); the differences between different energy statistics demonstrates that trends in $CO_2$ emissions are good indicators of trends in energy consumption. $NO_x$ and $SO_2$ also show marked differences in emission growth - increased by 197% and 45.1% during 1996-2012 according to the provincial energy statistics but by only 139% and 7.2% as determined from the original national energy statistics. From 2012 to 2013, the total energy consumption and $CO_2$

emissions, respectively, increased by 3.7% and 3.7% as seen from CT-CESY-3C but decreased by 0.4% and 2.1% according to PBP-CESY. The GDP increased by 7.7% between 2012 and 2013; thus, the decreasing trend in $CO_2$ emissions indicated by PBP-CESY is unexpected. Korsbakken et al. (2016) also pointed out that initial claims that Chinese $CO_2$ emissions fell in 2014 according to preliminary official statistics were probably premature. The unexpected energy and $CO_2$ emission decline



in 2013 in PBP-CESY could be explained by the fact that the PBP-CESY data for 2013, which probably include updates based on the third economic census, are closer to the data from CT-CESY-3C. As a result, the total growth rates since 1996 indicated by PBP-CESY and CT-CESY-3C are more similar to each other for the period 1996-2013 than periods before 2013 (e.g., 1996-2012). As part of the Chinese Five Year Plan, the Chinese government established a set of targets for emission reduction, including a 10% $SO_2$ reduction from the 2005 levels by 2010 and reductions of 10% in $NO_x$ and 8% in $SO_2$ from the 2010 levels by 2015. Our results show that because uncertainties in energy statistics can lead to the inference of different emission trends, reliable energy data are crucially important for obtaining accurate estimates of both the absolute levels of emissions and their trends.

## 4 Discussion

### 4.1 Understanding the reliability of energy statistics

The large uncertainties between the national and provincial energy statistics can be explained in terms of both inadequacies in China's statistical system and artificial factors. First, China's statistical data are generally collected and reported from bottom to top, and there is a lack of effective means of cross-checking at the local level; thus, these data are faced with problems such as data inconsistency and double counting (Wang et al., 2014). Inconsistencies between interprovincial imports and exports have been found in the provincial energy statistics—the sum of coal "interprovincial imports" is higher than the sum of coal "interprovincial exports" (Zhang et al., 2007). Also, provincial statistics more likely include double counting because certain interprovincial activities are claimed by all provinces involved. A similar situation affects economic statistics: the aggregate provincial GDP in 2012 is approximately 11% larger than the national total. Second, unlike for large- and medium-size enterprises, which have defined data collection and reporting procedures, the energy data for small enterprises are merely estimated, which could strongly degrade the data quality of the energy statistics (Jiang et al., 2009; Wang et al., 2011; Wang et al., 2014). A typical example is that the original official statistics (CT-CESY-Ori) did not fully count the coal production of illegal small coal mines, leading to underestimations in coal production around 2000 (Wang et al., 2011; Guan et al., 2012). Third, although there is no ample evidence of such activity, certain signs indicate that energy data may be modified for artificial purposes (Guan et al., 2012). The energy revisions after the second economic census (CT-CESY-2C) were found to bring the country closer to achieving its energy conservation targets (Aden et al., 2010). We also notice that some provinces had zero statistical difference, i.e., the supply data matches the consumption data exactly, which might mean that some provincial data were adjusted to achieve the exact match.

We compared China's coal consumption in 1996-2013 as indicated by different energy statistics from the supply perspective, as shown in Fig. 5. In the supply approach, energy consumption is estimated based on production, trade and changes in stock (consumption = production - exports + imports + change in stock + statistical difference). From the supply perspective, the national energy statistics tend to estimate conservative production and thus underestimate coal consumption. The coal



consumption data for 1996-2012 from the national energy statistics were revised upward after the first (CT-CESY-1C), second (CT-CESY-2C) and third (CT-CESY-3C) censuses because of increasing coal production, which may be largely explained by the small coal mines that were initially unaccounted for in the official statistics (Guan et al., 2012). Meanwhile, inconsistencies in interprovincial transport manifest as interprovincial net imports (see Fig. 5), resulting in a higher coal

supply in the provincial energy statistics, implying that either coal production is underestimated or coal consumption is overestimated. For the years before 2008, the coal production indicated by the provincial energy statistics is reasonably consistent with that derived from the original national energy statistics (CT-CESY-Ori) and lower than that from the revised national energy statistics (CT-CESY-3C), which could help partially explain the interprovincial net imports during this period as underestimates in production. Moreover, the provincial energy statistics likely include double counting and thus

might result in overestimates. This effect may be more significant in recent years with the more frequent collaboration among companies at the provincial level.

Satellite observations, which have been widely used in the assessment of emission trends in previous studies (e.g., Richter et al., 2005; van de A et al., 2008; Stavrakou et al., 2008; Lamsal et al., 2011), could be used as one independent approach to verifying energy statistics. Akimoto et al. (2006) compared trends in bottom-up $NO_x$ emissions with satellite-derived $NO_2$

columns for the period of 1996–2002 and found that the emission trends derived from various energy statistics were all lower than that inferred from the satellite observations (which increased by 50%). The PBP-CESY trends were within the uncertainty of the satellite observations, whereas the IEA2004 and CT-CESY trends were apparently underestimated beyond the uncertainty of the satellite observations. Zhang et al. (2007) compared trends over China in bottom-up $NO_x$ emissions with satellite $NO_2$ columns observed from 1996 to 2004 and found a larger trend in the satellite $NO_2$ columns than in the

$NO_x$ emissions. Berezin et al. (2013) derived top-down estimates of $CO_2$ emission trends by means of the satellite-derived $NO_x$ emission trends obtained using an inverse model and $CO_2$-to-$NO_x$ emission ratios (i.e., $CO_2/NO_x$) from bottom-up inventories. They also found a significant quantitative difference between bottom-up and indirect top-down estimates of the $CO_2$ emission trend for the period of 1996-2001, and the difference for the period of 2001-2008 was found to be in the range of possible systematic uncertainties associated with their estimation method.

Previous studies have investigated the decline in energy consumption between 1997 and 2001 indicated by CT-CESY-Ori (Akimoto et al., 2006; Sinton et al, 2000, 2001; Zhang et al., 2007; Berezin et al., 2013). This supposed decline was completely eliminated after the revisions following the three censuses. This fact may support the conclusion of these early studies that the trend in energy consumption indicated by the provincial energy statistics was more accurate than that derived from the unrevised national energy statistics during the early period. Liu et al. (2015) estimated total Chinese energy

consumption by adopting the apparent consumption approach and estimated a value for 2000-2012 that was 10 percent higher than that reported in China's national statistics before the third economic census and lower than that from the provincial energy statistics. The discrepancies between the national and provincial energy statistics were reduced after the three economic censuses. These facts indicate that the newest energy statistics after the third economic census may include



more cross-checks to reduce inconsistencies between the national and provincial energy statistics and thus can be recommended for use. In contrast, the energy consumption indicated by the national energy statistics from before the third economic census may be underestimated because of underestimations in energy production, whereas the energy consumption indicated by the provincial energy statistics from before the third economic census may be overestimated because of double
counting.

## 4.2 Implications for other studies

In this study, we find that uncertainties in energy statistics have great impacts on China's emission estimates, which could also be used to partially explain different emission estimates from other inventories. The Ministry of Environmental Protection of China (MEP) tends to estimate lower $SO_2$ and $NO_x$ emissions than MEIC (e.g., 30% lower for $SO_2$, and 20%
lower for $NO_x$ in 2012). Lower energy consumption from the national energy statistics, compared with provincial energy statistics, could help to explain the differences in emissions. Trends in $CO_2$ emissions are good indicators of trends in energy consumption, which can reflect the differences in energy statistics between different inventories. We compared the emission trends for $CO_2$ and $NO_x$ in MEIC, the Asian inventory REAS, and the global inventory EDGAR (Fig. 6). From 1996 to 2001, $CO_2$ emissions increased by 9.6% and 9.2% according to MEIC and REAS, respectively, but increased by only 0.4%
according to EDGAR; total energy consumption increased by 11.5% and 3.1% according to PBP-CESY and CT-CESY-1C, respectively, during the same period. From 1996 to 2008, $CO_2$ emissions increased by 135% according to MEIC but increased by only 114% according to EDGAR; total energy consumption increased by 138% and 110% according to PBP-CESY and CT-CESY-1C, respectively. These differences indicate that the energy consumption indicated by EDGAR, which was created using the IEA energy statistics, is likely closer to the national energy statistics.

The differences in $NO_x$ emission trends could be partially explained by differences in the energy statistics. During the period of 1996-2008, $NO_x$ emissions increased by 127% according to MEIC but by only 76% according to EDGAR. If the $CO_2$ growth trend were to be replaced with that from MEIC while keeping the same $NO_x$-to-$CO_2$ ratios, a greater (93%) increase in $NO_x$ emissions would be found from EDGAR. However, significant differences also arise from the emission factors (i.e., the $NO_x$-to-$CO_2$ ratios): the overall $NO_x$-to-$CO_2$ ratio decreased by only 3.5% in MEIC for the 1996-2008 period but
decreased by 17.9% in EDGAR. Similar trends in the overall $NO_x$-to-$CO_2$ ratio are found between MEIC and REAS: it increased from 1996 to 2001, primarily driven by a faster growth rate of diesel consumption (46-51%), which has a higher $NO_x$-to-$CO_2$ ratio, compared with the growth rate of coal consumption (-15-7%), but it then decreased after 2004, primarily because of the implementation of $NO_x$ emission standards in the power and transportation sectors. It should be noted that EDGAR tends to estimate a much earlier and more rapid decline in $NO_x$ emission factors compared with those seen from
MEIC and REAS (see Fig. 6(c)), for which the underlying driving forces are difficult to understand. For example, the $NO_x$-to-$CO_2$ ratios in EDGAR began to decrease significantly for the power and transportation sectors in 1993 and 1990,



respectively (see Fig. 6(d)), earlier than the years of implementation of major control measures regarding $NO_x$ emissions in these sectors (1996 and 2001, respectively) (State Environmental Protection Administration of China (SEPA), 1996, 2001).

The $CO_2$-to-$NO_x$ emission ratios taken from bottom-up inventories could be an important potential source of error in top-down estimates of $CO_2$ emission trends based on satellite $NO_2$ columns. Berezin et al. (2013) used the emission ratios from
EDGAR and found an increase in $CO_2$ emissions of as high as 240% for the 1996-2008 period using the top-down approach, much larger than the trends observed in bottom-up inventories (e.g., 114% in EDGAR). These substantial differences should be attributable mainly to the rapidly increasing $CO_2$-to-$NO_x$ ratios in EDGAR. If we adopt the emission ratios from MEIC (including uncertainties), we find an increase of 147-197%, much closer to the values from bottom-up inventories. Although uncertainties still exist, these results indicate that the energy consumption from EDGAR, which is similar to CT-CESY-1C,
as well as energy consumption in 2008 from CT-CESY-2C, is likely to be underestimated.

Top-down estimates of the $CO_2$-to-$NO_x$ emission ratios could offer an alternative approach. Reuter et al. (2014) used top-down estimation methods and found that the $CO_2$-to-$NO_x$ emission ratio for the years 2003-2011 in East Asia had increased by $4.2\pm1.7\%$ $yr^{-1}$. They found a large positive trend in $CO_2$ emissions in East Asia ($9.8\pm1.7\%$ $yr^{-1}$) that exceeded the positive trend in $NO_x$ emissions ($5.8\pm0.9\%$ $yr^{-1}$). The MEIC inventory reports a larger $CO_2$ trend in China (10.4% $yr^{-1}$) during the
same period. Reuter et al. (2014) noted a considerably smaller $CO_2$ trend in EDGAR (6.9% $yr^{-1}$) compared with these top-down estimates; it appears that considering the possible underestimations in Chinese $CO_2$ trends in EDGAR due to uncertainties in energy statistics could help to explain this difference. The MEIC inventory reports a larger $NO_x$ trend in China (8.1% $yr^{-1}$) than that reported by Reuter et al. (2014) for East Asia, which is consistent with Wang et al. (2014), who also found a faster $NO_x$ growth rate in China (34%) compared with that in East Asia as a whole (25%) for 2005-2010.

Zhao et al. (2011) estimated the uncertainties (i.e., 95% confidence intervals around the central estimates) of Chinese total $SO_2$, $NO_x$, and $PM_{2.5}$ emissions in 2005 to be -14~13%, -13~37%, and -17%~54%, respectively. We found that the apparent uncertainty ratios arising from the 2012 energy statistics for $SO_2$ and $NO_x$ emissions could be as large as 30.0% and 16.4%, respectively, indicating the importance of energy statistics to Chinese emission estimates for recent years, especially for $SO_2$ and $NO_x$. Variations at energy consumption could be an important source of emission uncertainties for $SO_2$ and $NO_x$. For
VOC and $PM_{2.5}$, uncertainties in energy consumption act as a minor source due to emission contributions from non-energy activities and large uncertainties from emission factors.

## 5 Conclusions

This study analyzed the apparent uncertainties in China's energy statistics and the impacts on China's estimated emissions for the period 1990-2013. We found increasing apparent uncertainties in China's energy consumption during 2004-2012 and
converging uncertainties in 2013. Coal is the dominant type of energy contributing to these uncertainties, and coal use in the industrial sector in particular is highly uncertain. Owing to high uncertainties in the energy statistics, the apparent



uncertainty ratios for emissions in 2012 are as large as 30.0%, 16.4%, 7.7%, 9.2% and 15.6%, for $SO_2$, $NO_x$, VOC, $PM_{2.5}$ and $CO_2$, respectively. $SO_2$ was found to be the most sensitive to energy uncertainties because of its high contribution from industrial coal combustion. The calculated emission trends are also greatly affected by energy uncertainties - from 1996 to 2012, $CO_2$ and $NO_x$ emissions, respectively, increased by 191% and 197% according to the provincial energy statistics but

by only 145% and 139% as determined from the original national energy statistics. For $SO_2$ and $NO_x$, the energy-induced emission uncertainties are comparable to total uncertainties of emissions as estimated by previous studies, indicating variations at energy consumption could be an important source of emission uncertainties. The reliability of the energy statistics cannot yet be regarded as conclusive, but possible explanations for the discrepancies include inconsistencies in interprovincial energy transport, double counting in provincial energy consumption, and underestimates in energy production

from small mines. While large uncertainties are present in this study, it is of critical importance to reform the statistical system, and to introduce more cross-checks and independent methods to help to verify the quality of energy data and to reduce uncertainties in energy consumption as well as emissions.

*Acknowledgements.* This work was supported by China's National Basic Research Program (2014CB441301), the National Science Foundation of China (41222036), the National Key Technology R&D Program (2014BAC16B03 and

2014BAC21B02), the public welfare program of China's Ministry of Environmental Protection (201509014). This work is a contribution to the TransChina project funded by the Research Council of Norway (235523). We thank Glen Peters and Jan Ivar Korsbakken for their helpful suggestions. Qiang Zhang and Kebin He are supported by the Collaborative Innovation Center for Regional Environmental Quality.

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



**Table 1. The energy statistics for China used in this work.**

| Energy Statistics | Data Source | National/ Provincial Level | Description |
|---|---|---|---|
| CT-CESY-Ori | NBS | National | For each year from 1990 to 2013, the original edition of the national Energy Balance Sheets published in the CESY was used. |
| CT-CESY-1C | NBS | National | For 1999-2003, the revised edition of the national Energy Balance Sheets released after the first economic census (published in CESY2005) was used; for other years, the data were the same as in CT-CESY-Ori. |
| CT-CESY-2C | NBS | National | For 1996-2007, the revised edition of the national Energy Balance Sheets released after the second economic census (published in CESY2009) was used; for other years, the data were the same as in CT-CESY-1C. |
| CT-CESY-3C | NBS | National | For 2000-2012, the revised edition of the national Energy Balance Sheets released after the third economic census (published in CESY2014) was used; for other years, the data were the same as in CT-CESY-2C. |
| PBP-CESY | NBS | Provincial | The provincial Energy Balance Sheets for each year published in the CESY were used. |
| CT-IEA-2012 | IEA | National | China's energy statistics from the IEA World Energy Balances (2012 edition) were used. |

Note: CESY, China Energy Statistics Yearbook; NBS, National Bureau of Statistics; IEA, International Energy Agency. CESY2005, CESY2009 and CESY2014 denote the revised national energy data for the periods of 1999-2003, 1996-2007 and 2000-2012, which were
5   released after the first, second and third economic censuses, respectively.



**Table 2. Apparent uncertainties in China's emissions in 2012 by sector and energy type. The apparent uncertainties are expressed in units of Tg. The percentages shown in parentheses indicate the apparent uncertainty ratios. Note that the emission uncertainties shown here are only those associated with energy uncertainties.**

|  | $CO_2$ | $NO_x$ | $SO_2$ | $PM_{2.5}$ | VOC |
|---|---|---|---|---|---|
| Total | 1633 (15.6%) | 4.68 (16.4%) | 7.76 (30.0%) | 1.10 (9.2%) | 1.90 (7.7%) |
| Power | 90 (2.7%) | 0.31 (3.3%) | 0.25 (3.7%) | 0.02 (2.7%) | 0.00 (2.6%) |
| Industry | 1196 (23.5%) | 3.39 (34.7%) | 6.04 (38.3%) | 0.51 (8.6%) | 1.00 (6.2%) |
| Residential | 201 (16.0%) | 0.17 (15.6%) | 1.42 (46.0%) | 0.50 (11.0%) | 0.33 (5.3%) |
| Transportation | 149 (18.3%) | 0.81 (9.9%) | 0.05 (17.6%) | 0.06 (11.6%) | 0.58 (25.0%) |
| Coal | 1350 (18.8%) | 3.57 (19.7%) | 7.58 (32.8%) | 1.03 (32.9%) | 1.28 (44.3%) |
| Petroleum | 193 (19.0%) | 0.90 (10.5%) | 0.20 (23.8%) | 0.07 (12.0%) | 0.59 (25.1%) |
| NG | 52 (20.0%) | 0.05 (17.7%) | 0 | 0 | 0.003 (23.7%) |
| Other fuels | 47 (14.4%) | 0.16 (21.0%) | 0 | 0.001 (19.8%) | 0.02 (2.0%) |





**Table 3. Emission trends for China derived from different energy statistics (growth rate, %).**

| Energy Statistics | $CO_2$ | $NO_x$ | $SO_2$ | $PM_{2.5}$ | VOC | $CO_2$ | $NO_x$ | $SO_2$ | $PM_{2.5}$ | VOC |
|---|---|---|---|---|---|---|---|---|---|---|
| | | | 1996-2003 | | | | | 2004-2012 | | |
| CT-CESY-Ori | 22.9 | 38.0 | 14.3 | -1.6 | 29.5 | 70.8 | 48.4 | -18.8 | -11.2 | 45.4 |
| CT-CESY-1C | 25.6 | 40.8 | 17.9 | -0.9 | 29.9 | 70.8 | 48.4 | -18.8 | -11.2 | 45.4 |
| CT-CESY-2C | 34.1 | 43.1 | 28.9 | 2.5 | 35.2 | 62.9 | 44.1 | -23.7 | -12.7 | 42.5 |
| CT-CESY-3C | 35.8 | 44.5 | 30.2 | 3.5 | 36.2 | 75.9 | 54.6 | -10.6 | -9.9 | 45.7 |
| PBP-CESY | 35.5 | 47.5 | 28.0 | 1.9 | 47.8 | 91.8 | 77.6 | 1.6 | -4.9 | 53.7 |
| | | | 1996-2000 | | | | | 1996-2012 | | |
| CT-CESY-Ori | -5.4 | 8.5 | -13.7 | -11.9 | 9.2 | 145 | 139 | 7.2 | -8.5 | 108 |
| CT-CESY-1C | -0.2 | 13.6 | -6.6 | -10.2 | 10.2 | 145 | 139 | 7.2 | -8.5 | 108 |
| CT-CESY-2C | 6.3 | 15.5 | 2.5 | -7.4 | 14.6 | 149 | 136 | 10.1 | -6.7 | 112 |
| CT-CESY-3C | 3.2 | 13.0 | -2.3 | -7.7 | 13.2 | 172 | 157 | 30.9 | -2.6 | 119 |
| PBP-CESY | 5.4 | 12.4 | 1.4 | -8.7 | 14.7 | 191 | 197 | 45.1 | 0.7 | 130 |





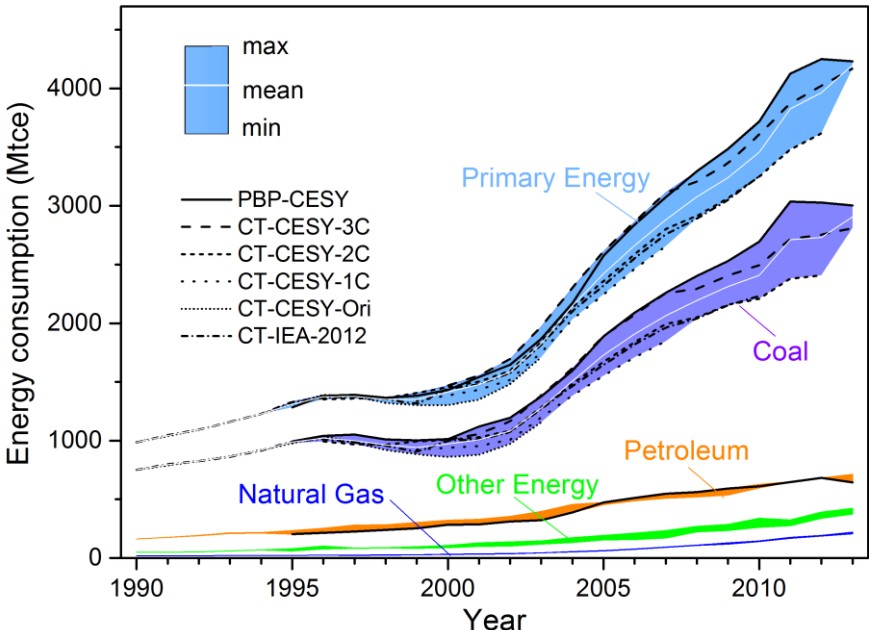

**Figure 1. Apparent uncertainties (shown as filled areas) in China's energy consumption from 1990 to 2013, by energy type. Note that CT-CESY-Ori, CT-CESY-1C, CT-CESY-2C and CT-CESY-3C are shown for 1990-2003, 1999-2007, 1996-2012 and 2000-2013, respectively.**





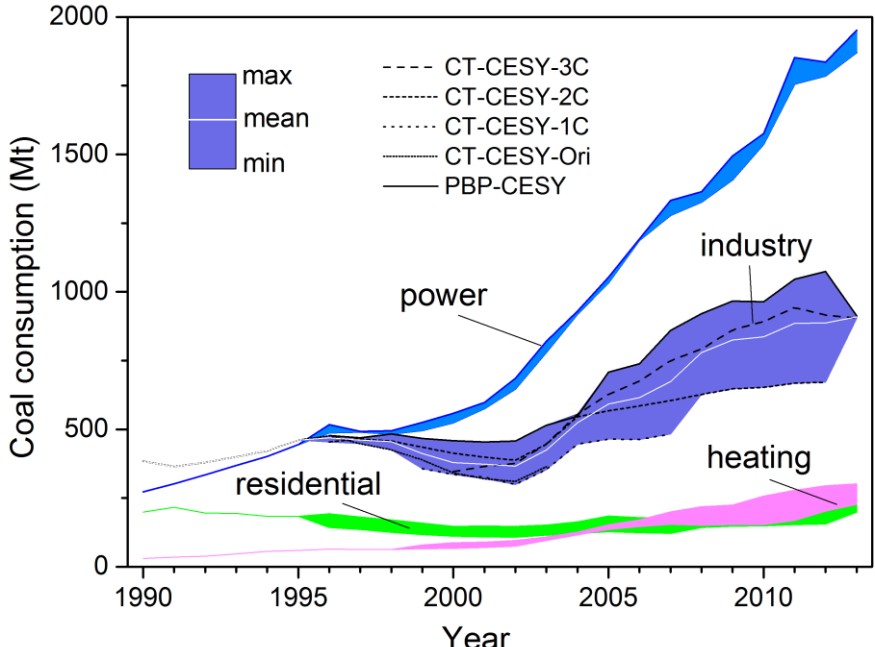

**Figure 2. Apparent uncertainties (shown as filled areas) in China's coal consumption from 1990 to 2013, by sector. Note that CT-CESY-Ori, CT-CESY-1C, CT-CESY-2C and CT-CESY-3C are shown for 1990-2003, 1999-2007, 1996-2012 and 2000-2013, respectively.**





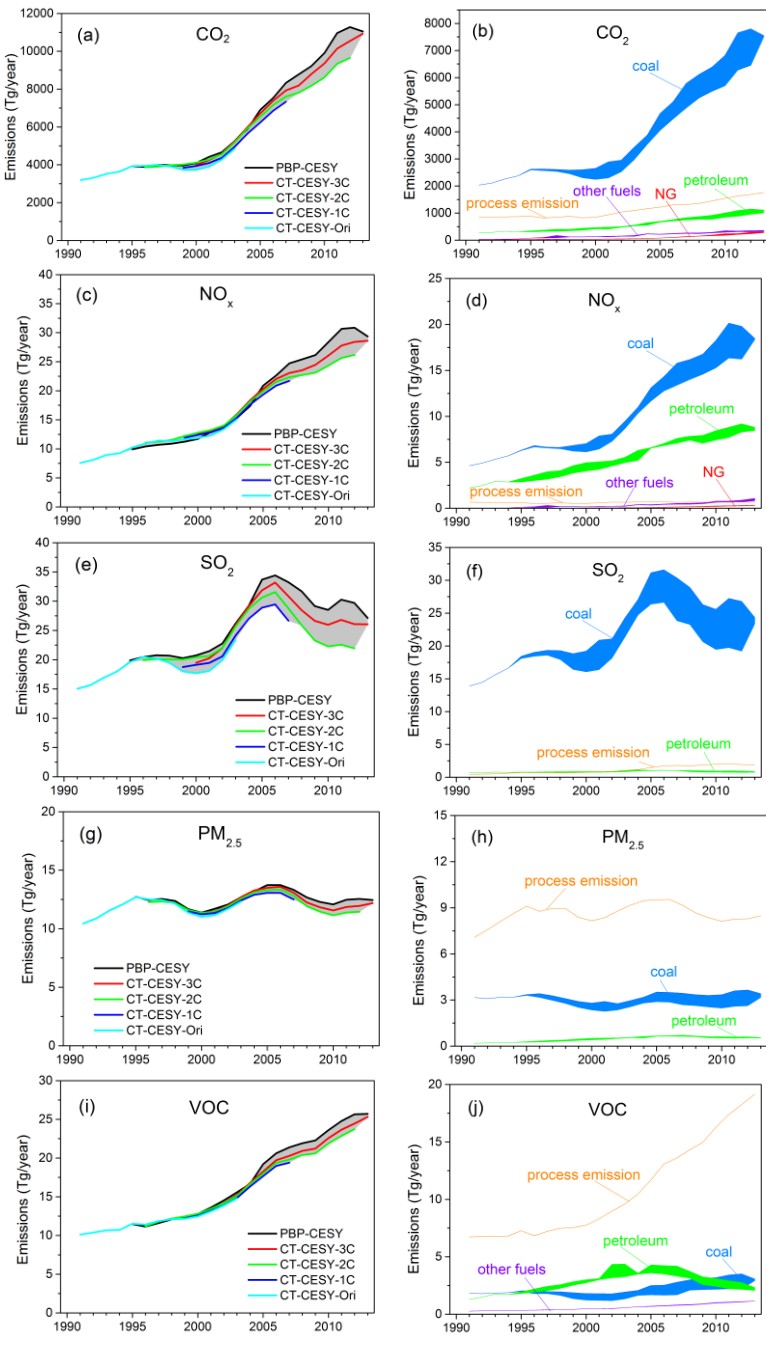

**Figure 3. Apparent uncertainties (shown as filled areas) in China's emissions during 1990-2013: (left) uncertainties in total emissions; (right) uncertainties by energy type. Note that the emission uncertainties shown here are only those associated with energy uncertainties. Note also that CT-CESY-Ori, CT-CESY-1C, CT-CESY-2C and CT-CESY-3C are shown for 1990-2003, 1999-2007, 1996-2012 and 2000-2013, respectively.**





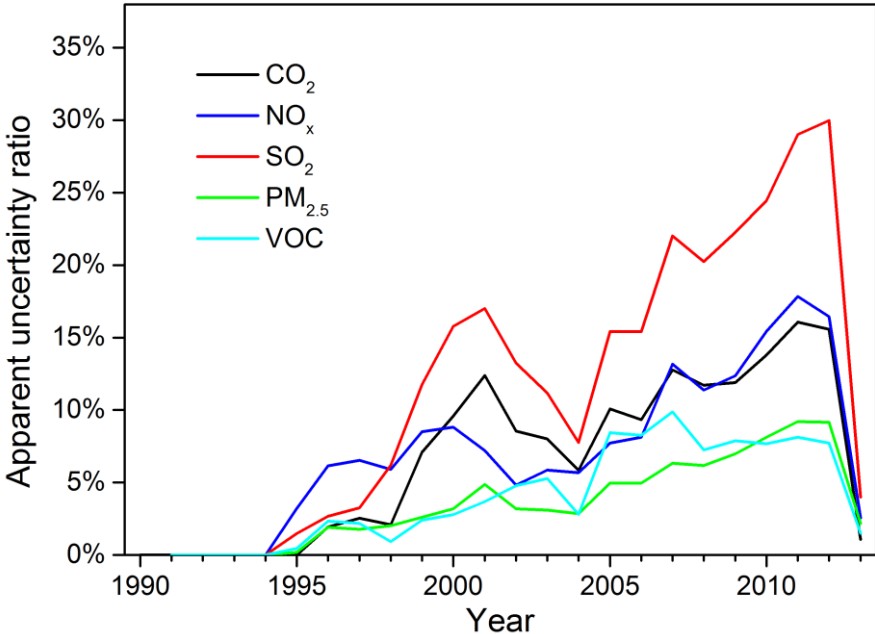

**Figure 4. Apparent uncertainty ratio in China's emissions during 1990-2013. Note that the emission uncertainties shown here are only those associated with energy uncertainties.**





5   **Figure 5. Differences in coal consumption between different energy statistics, from the supply perspective: (a) CT-CESY-1C minus CT-CESY-Ori; (b) CT-CESY-2C minus CT-CESY-1C; (c) CT-CESY-3C minus CT-CESY-2C; (d)PBP-CESY minus CT-CESY-Ori; (e) PBP-CESY minus CT-CESY-2C; and (f) PBP-CESY minus CT-CESY-3C. From the supply perspective, consumption = production - exports + imports + change in stock + statistical difference.**

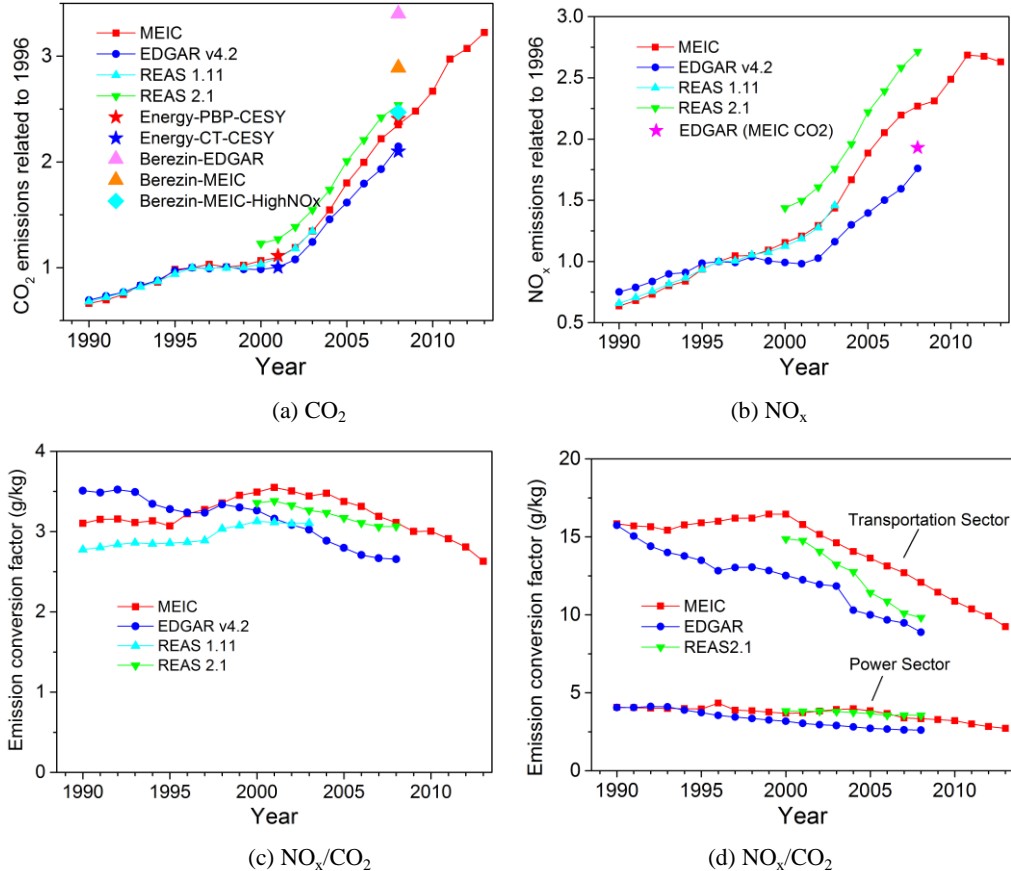

**Figure 6. (a) $CO_2$ and (b) $NO_x$ emission trends in China and the temporal evolution of the overall $NO_x$-to-$CO_2$ ratio (c) from different inventories and (d) in different major sectors according to the MEIC, EDGAR v4.2 and REAS (v1.11 and v2.1) emission inventories. The following trends (2008 values relative to 1996 values) are also shown: trends in total energy consumption from PBP-CESY (Energy-PBP-CESY) and CT-CESY-Ori (Energy-CT-CESY); $NO_x$ emission trends calculated using the $CO_2$ trend from MEIC and the $NO_x$-to-$CO_2$ ratios from EDGAR (EDGAR - MEIC $CO_2$); and satellite-derived $CO_2$ emission trends from Berezin et al. (2013) derived using the $CO_2$-to-$NO_x$ ratios from EDGAR (Berezin-EDGAR), MEIC (Berezin-MEIC) and the lower boundary of MEIC (Berezin-MEIC-High$NO_x$).**