# Peer review of "Variations of China's emission estimates response to uncertainties in energy statistics"

_Atmospheric Chemistry and Physics, 2016_

## Referee Comment (RC1) · Anonymous Referee #1 · 29 Jul 2016

The authors try to understand the uncertainties in China's energy statistics and estimate their impacts on China's emissions during the period of 1990-2013 using MEIC. The uncertainty of energy consumption statistics in China were appointed out in the previous many studies. This work is highly motivated and the authors try to understand the uncertainties of national statistics from China inside. Particularly, the discussion in the section '4.1 Understanding the reliability of energy statistics' is important. And also the authors analysis the uncertainties of emission inventory for air pollutants in China caused by those of energy statistics. In my knowledge, this is a first work. Additionally, the manuscript indicates the variations at energy consumption could be an important source of energy-induced emission uncertainties in China. The topic certainly is suitable for ACP. The authors define the apparent uncertainty as the maximum discrepancy among different datasets of energy consumption. My question is that this definition is

appropriate. For example, the converging in 2013 may be caused by any artificial modification because the trajectories in two datasets during 2010-2013 are quite different. This converging indicates small uncertainty? I think the "uncertainty" is unsuitable term and should be replaced to another term, such as "discrepancy in statistics" or other. In conclusion, the reviewer is recommending the minor revision of the manuscript.

———————————————————

---

## Referee Comment (RC2) · Anonymous Referee #2 · 22 Sep 2016

I have read the paper "Variations of China's emission estimates response to uncertainties in energy statistics" by Hong et al. This is a well structured study that addresses an important issue. The paper will be a solid addition to the literature. The paper should have a bit more background material along with some additional methodological details, as discussed below.

The sectoral resolution of the different datasets should be briefly discussed. First, is the sectoral resolution similar across all the datasets? I assume these data all distinguish between key sectors that have quite different emission factors - if so this should be stated? (e.g. iron and steel, vs boilers in industry; agricultural machinery vs road vehicles, etc.). If the sectoral resolution of the different datasets is not the same, then how this was treated in the data processing needs to be discussed (at least for sectors that are a signifiant portion of total emissions of one of the targeted species).

More details on the processing of the energy data are needed. All the text says now is "five emission inventories based on different sets of energy statistics (i.e., CT-CESY-Ori, CT-CESY-1C, CT-CESY-2C, CT-CESY-3C and PBP-CESY) were established.". In general, energy data sets do not contain all the information needed for an inventory, so additional assumptions (such as technology splits over time and technology retirements) would likely need to be made. Some assumptions likely had to be applied at some point, and these need to be described. Basically, the process of going from the energy datasets to the data needed in MEIC needs to be described. Then how this methodology might impact the results should be discussed. (If there are differences in the sectoral resolution of the different datasets, this could be an additional source of uncertainty, for example.)

Also, where fuel consumption differs between the datasets, how was this mapped to the technology detail in the inventory? For example, were the same emission factors applied for fuel consumption in a given sector in each year (even though different fuel consumption data would imply different rates of new purchases and/or retirements). Greater growth in coal consumption in one dataset as compared to another would tend to imply a greater amount of new equipment, which could have different emission factors as compared to older equipment. Note also that these assumptions would likely add additional uncertainty.

My understanding is the MEIC has province level detail. Were these calculations performed with province-specific emission factors, or national average emission factors. If the former, how were differences in national data allocated to provinces?

It would be useful to see a bit of a discussion of how these apparent uncertainties might extend back further in time. One point in particular, it should be noted that the narrowing of the uncertainty toward 1995 is due, in part, due to fewer different datasets. Can it be presumed that the methodologies for data collection did not evolve as much during this earlier period as compared to the latter statistical surveys (in which methodologies apparently became more consistent between provincial and national statistics)?

For this reason, I like the author's choice of terminology of "apparent uncertainty", but this possible bias in the results – e.g., actual uncertainty earlier in the series shown is likely be underestimated due to lack of multiple datasets – should be more explicitly discussed in the paper.

It would be useful if the authors could discuss a bit more possible reasons why the provincial and national statistics agree during earlier time periods. Was this because both of these statistics contained similar biases? Or were there some potential sources of bias that increased over this time period. The authors have substantial experience with these datasets and their insights (although likely no firm answers!) into these issues, and a more complete discussion would greatly strength and add to the value of this paper.

SPECIFIC COMMENTS In Table 1" is described as "The energy statistics for China used in this work." and IEA data are included in this table. However, the text states that "The IEA energy statistics were excluded from the emission calculations because they are based on NBS's national Energy Balance Sheets". Please clarify (I believe it is useful to have IEA data in Table 1, since it gives context for this widely used dataset, but perhaps add a footnote that these data are not used in the current work, or re-title the table.)

Page 6, line 31 "contributions (approximately 70%) from industrial process emissions". It would be useful to clarify by adding (I assume this is the case) "contributions (approximately 70%) from industrial process emissions. Note that non-combustion emission uncertainty was not addressed in this study."

This brings up an additional point. Were all fuel consumption differences assumed to be applied to combustion sectors? Or was some portion of these differences assumed to be feedstocks? This should be clarified in the paper.

Page 7, line 7 "The contributions of gas and other fuels are negligible because their emissions are relatively small." This is not necessarily true for biomass (which often

contributes substantially to CO emissions in particular). I assume that uncertainty in biomass consumption was not included in this study? If uncertainty in biomass consumption was not considered this would be useful to state here (and also needs to be mentioned earlier in the methodology section).

Page 8, line 23 "Third, although there is no ample evidence of such activity" ample is not quite the correct word to use here (is ambiguous). Depending on what the authors mean, a clearer words should be used.

Page 11, line 11 "Top-down estimates of the CO2-to-NOx emission ratios". Give the reader a short definition of how top-down differs from bottom up. Presumably this is observationally based?

Page 11, line 14. "The MEIC inventory reports a larger CO2 trend in China (10.4% yr-1) " it looks like this is not larger, it is well within the uncertainty of the top-down estimate.

page 11, conclusion section Re-define "apparent uncertainty" here so that the conclusion is more easily understood on its own.

Figure 5 is a bit difficult to interpret due to the many different parings of inventories. The authors might want to experiment to see if a consistent set of differences (e.g. showing the difference between each dataset vs one dataset that spans all years (if available) would communicate the points they wish to make, so that there is a consistent reference over the entire period). This might be more straightforward for the readers to interpret.

---

## Referee Comment (RC3) · Anonymous Referee #3 · 28 Sep 2016

Overall Quality

Although this paper seeks to address an important topic and is well written, it lacks sufficient scientific merit for publication. It repeats the general strategy of an earlier paper by one of the co-authors (Guan et al. 2012) that sought to repackage the existence of large inconsistencies between different official Chinese datasets concerning energy as an analytical research finding. Those inconsistencies are important to understanding China's air pollution and greenhouse gas emissions, but their existence is not newly recognized (Sinton 2001; Akimoto et al. 2006) and, more importantly, processing them with pre-packaged emission estimation protocols does not yield findings that should be considered publishable original research.

The paper essentially does the following. First, it assembles a set of publicly available

energy datasets. Second, it processes these datasets using the pre-existing MEIC model for calculating atmospheric emissions. And third, it uses statistically questionable comparisons of the resulting disparities in both energy and emission datasets to draw inferences about the scale and sources of emission uncertainty. The mechanical processing of existing datasets using preexisting (and opaque) research tools is neither innovative nor novel. Importantly, it is also not reproducible, at least as currently presented. Last, the inferences about uncertainty are speculative, as no rationales for use of the metrics defined and employed in the paper are presented.

The extent to which the results are interesting is derived from the scale of the inconsistencies of the underlying data, not from the analysis itself. While the authors appear positioned to undertake a more rigorous assessment of their important topic, the current paper is too formulaic, unsupported, and speculative to justify publication.

Individual Questions/Issues

1. The paper is irreproducible, as it does not describe the methods of estimating emissions applied to different energy consumption datasets. It instead refers the reader to the website of the MEIC model, which does not present all of the underlying data and assumptions of the emission estimation model. To be reproducible, methods and assumptions must be described for each category of energy use (industrial subsector, for example, or vehicle type) treated uniquely in the assessment. Other researchers therefore cannot replicate the emission estimation as currently presented, except by blind trust in the same MEIC model.

2. The paper draws inferences about uncertainty based on two values defined in lines 9-10 of page 3: "We defined the apparent uncertainty as the maximum discrepancy among different datasets and the apparent uncertainty ratio as the ratio of the maximum discrepancy to the mean value from the different datasets." These two concepts sound attractive but the rationales for their use to draw inferences about statistical uncertainty are currently lacking in the paper.

Both concepts appear problematic. Regarding "apparent uncertainty," a case has been made that a rough estimate of the uncertainty of energy data might be based on differences in values of subsequently revised data in the same official series (Marland et al. 2009). The rationale rests on a reasonable expectation that revisions represent increasing accuracy in the data and/or calculations, or "learning and convergence." In the current paper, however, any connection to this rationale is lost because the authors simply compile datasets from different series (national, provincial, and IEA) and seek a maximum differential. Some sort of conceptual rationale for readers to find meaning in the value defined as apparent uncertainty is required for this calculation to be interpretable.

The "apparent uncertainty ratio" is problematic first because the numerator is apparent uncertainty, with the conceptual concern just noted, but then compounded by a denominator that is also hard to rationalize because of autocorrelation. Taking the mean of all datasets, including sequential revisions of the same dataset (CT-CESY-Ori, CT-CESY-1C, CT-CESY-2C, and CT-CESY-3C), implicitly assumes that they are independent. Without a defensible justification of this assumption, the calculations should recognize that revisions represent improving accuracy and should not be treated equally (as in a mean) in assessment of uncertainty.

The paper requires a more rigorously conceived statistical basis to draw the sort of inferences about uncertainty that it seeks as its primary conclusions.

Technical Corrections:

The authors need to revisit the above fundamental issues first before they (and reviewers) put time into other issues and technical corrections that this paper needs.

References:

Citations are referenced in the manuscript except:

Marland, G., Hamal, K., and Jonas, M.: How uncertain are estimates of $CO_2$ emissions?, J. Ind. Ecol., 13, 4–7, 2009.

---

## Author Comment (AC1) · 14 Nov 2016

*The authors try to understand the uncertainties in China's energy statistics and estimate their impacts on China's emissions during the period of 1990-2013 using MEIC. The uncertainty of energy consumption statistics in China were appointed out in the previous many studies. This work is highly motivated and the authors try to understand the uncertainties of national statistics from China inside. Particularly, the discussion in the section '4.1 Understanding the reliability of energy statistics' is important. And also the authors analysis the uncertainties of emission inventory for air pollutants in China caused by those of energy statistics. In my knowledge, this is a first work. Additionally, the manuscript indicates the variations at energy consumption could be an important source of energy-induced emission uncertainties in China. The topic certainly is suitable for ACP.*

**Response:** We thank Referee #1 for the encouraging comments. We address the comments as below.

*The authors define the apparent uncertainty as the maximum discrepancy among different datasets of energy consumption. My question is that this definition is appropriate. For example, the converging in 2013 may be caused by any artificial modification because the trajectories in two datasets during 2010-2013 are quite different. This converging indicates small uncertainty? I think the "uncertainty" is unsuitable term and should be replaced to another term, such as "discrepancy in statistics" or other. In conclusion, the reviewer is recommending the minor revision of the manuscript.*

**Response:** We agree with the referee that the terminology of "apparent uncertainty" may make some confusion. In the introduction section of the revised manuscript, in order to avoid confusion, we have clarified the meaning of "apparent uncertainty" defined in this study as compared to the meaning of "actual uncertainty". Apparent uncertainty is a straightforward metric used to quantitatively gauge the apparent discrepancies between different existing datasets. Apparent uncertainty ratio is a metric to quantify the relative deviation. Thus apparent uncertainty could partly reflect actual uncertainty. In general, large apparent uncertainty reflects large discrepancies, which might indicate large actual uncertainty. However, it should be noted that apparent uncertainty could not fully represent actual uncertainty, and apparent uncertainty would likely to be conservative estimates as it might be subjected to the datasets used. Thus small apparent uncertainty does not necessarily mean to small actual uncertainty. For example, the small apparent uncertainties before 1996 might become larger if a new energy dataset that revises the data of this period is included. The converging apparent uncertainties in 2013 may be caused by the third economic census. We have clarified this in the Section 3.2 of the revised manuscript.

---

## Author Comment (AC2) · 14 Nov 2016

*I have read the paper "Variations of China's emission estimates response to uncertainties in energy statistics" by Hong et al. This is a well structured study that addresses an important issue. The paper will be a solid addition to the literature. The paper should have a bit more background material along with some additional methodological details, as discussed below.*

**Response:** We thank Referee #2 for the constructive comments. We address the comments as below.

*The sectoral resolution of the different datasets should be briefly discussed. First, is the sectoral resolution similar across all the datasets? I assume these data all distinguish between key sectors that have quite different emission factors - if so this should be stated? (e.g. iron and steel, vs boilers in industry; agricultural machinery vs road vehicles, etc.). If the sectoral resolution of the different datasets is not the same, then how this was treated in the data processing needs to be discussed (at least for sectors that are a signifiant portion of total emissions of one of the targeted species).*

**Response:** The sectoral categories are consistent across all the energy datasets from NBS. In the supplement of the revised manuscript, we provide the sector information of the NBS energy statistics. The same scale factor in fuel consumption was applied for all the sub-categories in same major sector (e.g. industrial coal-fired boilers and kilns in industry sector; on-road diesel vehicles and off-road mobile sources in transportation sector). We have clarified this in the Section 2.2 of the revised manuscript.

*More details on the processing of the energy data are needed. All the text says now is "five emission inventories based on different sets of energy statistics (i.e., CT-CESYOri, CT-CESY-1C, CT-CESY-2C, CT-CESY-3C and PBP-CESY) were established.". In general, energy data sets do not contain all the information needed for an inventory, so additional assumptions (such as technology splits over time and technology retirements) would likely need to be made. Some assumptions likely had to be applied at some point, and these need to be described. Basically, the process of going from the energy datasets to the data needed in MEIC needs to be described. Then how this methodology might impact the results should be discussed. (If there are differences in the sectoral resolution of the different datasets, this could be an additional source of uncertainty, for example.)*

**Response:** The emissions in MEIC were estimated as a product of the activity rate (such as energy consumption or material production), the technology distributions of fuel/production and emission control, the unabated emission factor, and the removal efficiency. Thus, the emission estimates can be simplified as the activity rates multiplied by their respective net emission factors of different fuel/product types in different sectors. Note that the net emission factors in MEIC change dynamically driven by the technology renewal process year by year. Technology distributions within each sector are obtained from Chinese statistics, a wide range of unpublished statistics by various industrial association and technology reports. For example, technology distributions in the power sector were obtained based on unit-base database (Liu et al., 2015). Technology distributions in the transportation sector were estimated based on fleet model (Zheng et al., 2014). The methods on emission estimates has been documented in our previous work (Zhang et al., 2007; Zhang et al., 2009; Zheng et al., 2014; Liu et al., 2015). We have described this in Section 2.2 of the revised manuscript.

*Also, where fuel consumption differs between the datasets, how was this mapped to the technology*

*detail in the inventory? For example, were the same emission factors applied for fuel consumption in a given sector in each year (even though different fuel consumption data would imply different rates of new purchases and/or retirements). Greater growth in coal consumption in one dataset as compared to another would tend to imply a greater amount of new equipment, which could have different emission factors as compared to older equipment. Note also that these assumptions would likely add additional uncertainty.*

**Response:** For different energy datasets, the same net emission factors were applied for fuel consumption in a given sector in each year during the emission calculations. MEIC already simulates the dynamic changes in net emission factors driven by the technology renewal process year by year. In fact, energy differences might change the technology renewal process, and further change the net emission factors. However, considering that those assumptions would likely add additional uncertainty and we do not discuss the uncertainties in emission factors, such indirect impacts on emission factors are not included in this study. We have clarified this in the Section 2.2 of the revised manuscript.

*My understanding is the MEIC has province level detail. Were these calculations performed with province-specific emission factors, or national average emission factors. If the former, how were differences in national data allocated to provinces?*

**Response:** As the emission calculations were performed with province-level data, energy consumption in the national energy statistics were directly allocated to provinces by using the ratios derived from the provincial energy statistics. We have clarified this in the Section 2.2 of the revised manuscript.

*It would be useful to see a bit of a discussion of how these apparent uncertainties might extend back further in time. One point in particular, it should be noted that the narrowing of the uncertainty toward 1995 is due, in part, due to fewer different datasets. Can it be presumed that the methodologies for data collection did not evolve as much during this earlier period as compared to the latter statistical surveys (in which methodologies apparently became more consistent between provincial and national statistics)?*

**Response:** In the Section 3.2 of the revised manuscript, we have added a paragraph to discuss how these apparent uncertainties extend in time. It should be noted that the apparent uncertainties calculated in this study are subjected to the energy datasets used. For example, the small apparent uncertainties before 1996 might become larger if a new energy dataset that revises the data of this period is included. Apparent uncertainties during the recent period of rapid growth (2004-2012) are higher than the early period (1996-2003), implying that the discrepancies might be accumulated and expanded for a period of rapid growth. For example, underestimates of the growth trends of small enterprises might result into accumulated underestimations. Note that the energy consumption apparently became more consistent between provincial and national statistics after the three economic censuses, indicating that the new energy statistics after the economic census may include evolved methodologies for data collection and more cross-checks to reduce the discrepancies. In this case, conducting censuses in some interval years could help to reduce the accumulated discrepancies. The apparent uncertainty ratio in years economic censuses newly conducted (i.e., 2004, 2008 and 2013) is generally less than that of previous years (i.e., 2003, 2007 and 2012), as shown in Figure 4. The converging uncertainties in 2013 may also be caused by the third economic census.

*For this reason, I like the author's choice of terminology of "apparent uncertainty", but this possible*

*bias in the results – e.g., actual uncertainty earlier in the series shown is likely be underestimated due to lack of multiple datasets – should be more explicitly discussed in the paper.*

**Response:** In the introduction section of the revised manuscript, in order to avoid confusion, we have clarified the meaning of "apparent uncertainty" defined in this study as compared to the meaning of "actual uncertainty". Apparent uncertainty is a straightforward metric used to quantitatively gauge the apparent discrepancies between different existing datasets. Apparent uncertainty ratio is a metric to quantify the relative deviation. Thus apparent uncertainty could partly reflect actual uncertainty. In general, large apparent uncertainty reflects large discrepancies, which might indicate large actual uncertainty. However, it should be noted that apparent uncertainty could not fully represent actual uncertainty, and apparent uncertainty would likely to be conservative estimates as it might be subjected to the datasets used. Thus small apparent uncertainty does not necessarily mean to small actual uncertainty. For example, the small apparent uncertainties before 1996 might become larger if a new energy dataset that revises the data of this period is included. We have clarified this in the Section 3.2 of the revised manuscript.

*It would be useful if the authors could discuss a bit more possible reasons why the provincial and national statistics agree during earlier time periods. Was this because both of these statistics contained similar biases? Or were there some potential sources of bias that increased over this time period. The authors have substantial experience with these datasets and their insights (although likely no firm answers!) into these issues, and a more complete discussion would greatly strength and add to the value of this paper.*

**Response:** Apparent uncertainties during the recent period of rapid growth (2004-2012) are higher than the early period (1996-2003), implying that the discrepancies might be accumulated and expanded for a period of rapid growth. For example, underestimates of the growth trends of small enterprises might result into accumulated underestimations. In this case, conducting censuses in some interval years could help to reduce the accumulated discrepancies. We have discussed this in the Section 3.2 of the revised manuscript.

SPECIFIC COMMENTS

*In Table 1" is described as "The energy statistics for China used in this work." and IEA data are included in this table. However, the text states that "The IEA energy statistics were excluded from the emission calculations because they are based on NBS's national Energy Balance Sheets". Please clarify (I believe it is useful to have IEA data in Table 1, since it gives context for this widely used dataset, but perhaps add a footnote that these data are not used in the current work, or re-title the table.)*

**Response:** We have changed the title of Table 1 as "The energy statistics for China involved in this work.", and add a footnote that the IEA energy statistics were used for comparison, but they were excluded from the uncertainty calculations in the current work. The IEA energy statistics are generally based on NBS's national Energy Balance Sheets, and currently quite consistent with CT-CESY-2C. They may soon be updated based on CT-CESY-3C. We have also changed the description in Section 2 accordingly.

*Page 6, line 31 "contributions (approximately 70%) from industrial process emissions". It would be useful to clarify by adding (I assume this is the case) "contributions (approximately 70%) from industrial process emissions. Note that non-combustion emission uncertainty was not addressed in this*

*study."*

**Response:** We have clarified this by adding "Note that non-combustion emission uncertainty was not addressed in this study." in the Section 3.2 of the revised manuscript.

*This brings up an additional point. Were all fuel consumption differences assumed to be applied to combustion sectors? Or was some portion of these differences assumed to be feedstocks? This should be clarified in the paper.*

**Response:** We only applied all the fuel consumption differences to the combustion sectors. In fact, differences in energy consumption would imply differences in feedstocks and products. However, the possible uncertainties in feedstocks and products resulted from energy uncertainties are not included in this study for some reasons. First, energy statistics and industrial products statistics in China are independent statistics. Inconsistencies may be existed between the energy data and the production data, and some studies used them for cross-checks (Guan et al., 2012; Korsbakken et al., 2016). Also, feedstocks and products usually have more detailed categories than energy sectors (e.g., iron and steel vs. industry sector). Thus, estimates of feedstocks and products based on energy data would introduce additional uncertainties. Without considering the possible uncertainties in feedstocks and products, our estimates of emission uncertainties are likely on the conservative side. We have clarified this in Section 2.2 of the revised manuscript.

*Page 7, line 7 "The contributions of gas and other fuels are negligible because their emissions are relatively small." This is not necessarily true for biomass (which often contributes substantially to CO emissions in particular). I assume that uncertainty in biomass consumption was not included in this study? If uncertainty in biomass consumption was not considered this would be useful to state here (and also needs to be mentioned earlier in the methodology section).*

**Response:** In the original manuscript, biomass emissions were put into "process emissions". In the revised manuscript, to make it be more straightforward, biomass emissions were moved to "other fuels", which also changed Figure 3 and Table 2. The contributions of gas and other fuels are negligible because uncertainties in biomass consumption are not included in this study and other emissions are relatively small. Note that biomass consumption, which is usually thought to be quite uncertain, would contribute more uncertainties in emissions. We have clarified this in the Section 2.2 and the Section 3.2 of the revised manuscript.

*Page 8, line 23 "Third, although there is no ample evidence of such activity" ample is not quite the correct word to use here (is ambiguous). Depending on what the authors mean, a clearer words should be used.*

**Response:** In order to avoid confusion, we have removed the sentence "although there is no ample evidence of such activity," in the revised manuscript.

*Page 11, line 11 "Top-down estimates of the CO2-to-NOx emission ratios". Give the reader a short definition of how top-down differs from bottom up. Presumably this is observationally based?*

**Response:** We have clarified this by adding "using satellite observations" in the revised manuscript.

*Page 11, line 14. "The MEIC inventory reports a larger CO2 trend in China (10.4% yr-1) " it looks like this is not larger, it is well within the uncertainty of the top-down estimate.*

**Response:** We have changed the word "larger" to "similar" in the revised manuscript.

*page 11, conclusion section Re-define "apparent uncertainty" here so that the conclusion is more easily understood on its own.*

**Response:** To re-define "apparent uncertainty" in the conclusion section of the revised manuscript, we have added the term "maximum discrepancy" after "apparent uncertainty", and the term "the ratio of the maximum discrepancy to the mean value" after "apparent uncertainty ratios".

*Figure 5 is a bit difficult to interpret due to the many different parings of inventories. The authors might want to experiment to see if a consistent set of differences (e.g. showing the difference between each dataset vs one dataset that spans all years (if available) would communicate the points they wish to make, so that there is a consistent reference over the entire period). This might be more straightforward for the readers to interpret.*

**Response:** We have combined different parings of the national statistics into one figure, and removed some figures. In the revised manuscript, Figure 5(a) compares different national statistics, showing that the coal consumption data from the national energy statistics were revised upward after the three censuses because of increasing coal production; Figure 5(b) shows that inconsistencies in interprovincial transport manifest as interprovincial net imports, resulting in a higher coal supply in the provincial energy statistics, implying that either coal production is underestimated or coal consumption is overestimated.

**Reference**

Guan, D. B., Liu, Z., Geng, Y., Lindner, S., and Hubacek, K.: The gigatonne gap in China's carbon dioxide inventories, Nat. Clim. Change, 2, 672–675, doi:10.1038/Nclimate1560, 2012.

Korsbakken, J. I., Peters, G. P., and Andrew, R. M.: Uncertainties around reductions in China's coal use and $CO_2$ emissions, Nat. Clim. Change, 1–5, doi:10.1038/ Nclimate 2963, 2016.

Liu, F., Zhang, Q., Tong, D., Zheng, B., Li, M., Huo, H., and He, K. B.: High-resolution inventory of technologies, activities, and emissions of coal-fired power plants in China from 1990 to 2010, Atmos. Chem. Phys., 15, 13299–13317, doi:10.5194/acp-15-13299-2015, 2015.

Zhang, Q., Streets, D. G., He, K., Wang, Y., Richter, A., Burrows, J. P., Uno, I., Jang, C. J., Chen, D., Yao, Z., and Lei, Y.: NOx emission trends for China, 1995-2004: The view from the ground and the view from space, J. Geophys. Res., 112, D22306, doi:10.1029/2007jd008684, 2007.

Zhang, Q., Streets, D. G., Carmichael, G. R., He, K. B., Huo, H., Kannari, A., Klimont, Z., Park, I. S., Reddy, S., Fu, J. S., Chen, D., Duan, L., Lei, Y., Wang, L. T., and Yao, Z. L.: Asian emissions in 2006 for the NASA INTEX-B mission, Atmos. Chem. Phys., 9, 5131–5153, 2009.

Zheng, B., Huo, H., Zhang, Q., Yao, Z. L., Wang, X. T., Yang, X. F., Liu, H., and He, K. B.: High-resolution mapping of vehicle emissions in China in 2008, Atmos. Chem. Phys., 14, 9787–9805, doi:10.5194/acp-14-9787-2014, 2014.

---

## Author Comment (AC3) · 14 Nov 2016

*Overall Quality*

*Although this paper seeks to address an important topic and is well written, it lacks sufficient scientific merit for publication. It repeats the general strategy of an earlier paper by one of the co-authors (Guan et al. 2012) that sought to repackage the existence of large inconsistencies between different official Chinese datasets concerning energy as an analytical research finding. Those inconsistencies are important to understanding China's air pollution and greenhouse gas emissions, but their existence is not newly recognized (Sinton 2001; Akimoto et al. 2006) and, more importantly, processing them with pre-packaged emission estimation protocols does not yield findings that should be considered publishable original research.*

**Response:** We thank Referee #3 for the constructive comments. Emission inventories over China are thought to be quite uncertain due to incomplete knowledge of activity rates and emission factors. For a long period, the emission inventory community assumed that the uncertainties in energy statistics are small and attributed the main sources of uncertainties to emission factors (Streets et al., 2003; Lu et al., 2011; Zhao et al., 2011; Kurokawa et al., 2013). For example, Zhao et al. (2011) assumed normal distributions with CV of 10% for energy consumption in the industry sector. Large differences among different statistics (Sinton 2001) and their impacts on emission estimates for $CO_2$ (Guan et al., 2012) and $NO_x$ (Akimoto et al., 2006) have been identified by previous studies, however, the impacts on the emission estimates of different air pollutants covering a long-term period were not well recognized from previous studies.

In this work, we evaluate the impacts on major air pollutants (i.e., $SO_2$, $NO_x$, VOC and $PM_{2.5}$) and include the recent energy statistics covering a full period (1990-2013). We found that the uncertainties induced by energy statistics are much higher than the assumptions in previous studies. For example, we identified that using different statistics could introduce as high as 30% differences in $SO_2$ emission estimates over China, which is larger than the previously estimated uncertainty range of $SO_2$ emissions in China (i.e., -14%~13% from Zhao et al., 2011). We also found increasing uncertainties in China's energy consumption during 2004-2012, and converging uncertainties in 2013. Our findings indicate that variations in energy statistics could be an important source of China's emission uncertainties. Given that, we believe that our study provides important and new findings on the knowledge of uncertainties in bottom-up emission inventories and merits publication in ACP.

Both referee #1 and #2 endorsed the novelty of our work. As indicated by Referee #1, "In my knowledge, this is a first work." As indicated by Referee #2, "The paper will be a solid addition to the literature." The results and methods presented in this study could be used to distinguish how many differences in emissions between two existing inventories might come from those inconsistencies in energy data, as presented in Section 4.2. We believe the paper will be helpful to improve understanding of East Asia emissions, and it is quite suitable for the special issue of "East Asia emissions assessment (EA2)".

*The paper essentially does the following. First, it assembles a set of publicly available energy datasets. Second, it processes these datasets using the pre-existing MEIC model for calculating atmospheric emissions. And third, it uses statistically questionable comparisons of the resulting disparities in both energy and emission datasets to draw inferences about the scale and sources of emission uncertainty. The mechanical processing of existing datasets using preexisting (and opaque) research tools is neither innovative nor novel. Importantly, it is also not reproducible, at least as currently presented. Last, the*

*inferences about uncertainty are speculative, as no rationales for use of the metrics defined and employed in the paper are presented.*

**Response:** We clarified the question about reproducibility here. For the question about the uncertainty metrics defined in this study, please refer to the following responses to "Individual Questions/Issues".

The MEIC emission inventory model (available at http://www.meicmodel.org) was used in this study to investigate the emission responses to different energy statistics. MEIC is a dynamic technology-based inventory developed by Tsinghua University. The methodology and data used in developing the MEIC model has been extensively documented in our previous publications, ensuring the transparency and reproducibility of the MEIC model (e.g., Zhang et al., 2007; Zhang et al., 2009; Lu et al., 2010; Lei et al., 2011; Zheng et al., 2014; Huo et al., 2015; Liu et al., 2015). The MEIC inventory has been widely used in supporting air quality models (e.g., Geng et al., 2015; Li et al., 2015; Zhang et al., 2015; Liu et al., 2016), and evaluated against surface and satellite-based observations (e.g., Chen et al., 2015; Zheng et al., 2015; Hu et al., 2016). To make the paper be more reproducible, we have provided more detailed methods in the Section 2.2 of the revised manuscript. The MEIC is developed following the work of INTEX-B (Zhang et al., 2009), with several updates, such as a unit-based emission inventory of power plants (Liu et al., 2015), a high-resolution vehicle emission inventory at the county level (Zheng et al., 2014), and an improved NMVOC speciation approach for various chemical mechanisms (Li et al., 2014). MEIC inventory includes recent control policies based on the available official reports (Ministry of Environmental Protection of China (MEP), 1991-2014, 2000-2014). The emissions in MEIC were estimated as a product of the activity rate (such as energy consumption or material production), the technology distributions of fuel/production and emission control, the unabated emission factor, and the removal efficiency. Thus, the emission estimates can be simplified as the activity rates multiplied by their respective net emission factors of different fuel/product types in different sectors. Note that the net emission factors in MEIC change dynamically driven by the technology renewal process year by year. Technology distributions within each sector are obtained from Chinese statistics, a wide range of unpublished statistics by various industrial association and technology reports. For example, technology distributions in the power sector were obtained based on unit-base database (Liu et al., 2015). Technology distributions in the transportation sector were estimated based on fleet model (Zheng et al., 2014). The methods on emission estimates has been documented in our previous work (Zhang et al., 2007; Zhang et al., 2009; Zheng et al., 2014; Liu et al., 2015).

The methods of estimating emissions applied to different energy consumption datasets have been clarified more carefully in the Section 2.2 of revised manuscript. To further explore the impact of energy data inconsistencies on estimates of China's emissions, five emission inventories based on five sets of energy statistics (i.e., CT-CESY-Ori, CT-CESY-1C, CT-CESY-2C, CT-CESY-3C and PBP-CESY) were established in the framework of the MEIC inventory. Note that only energy data were changed in the calculations of these emission inventories, while other data such as net emission factors remained the same as MEIC inventory. Thus the emission uncertainties derived from these inventories are only those associated with energy uncertainties. They do not include uncertainties in the emission factors and other parameters in MEIC inventory, which is not addressed in this study. For different energy datasets, the same net emission factors were applied for fuel consumption in a given sector in each year during the emission calculations. In fact, energy differences might change the technology renewal process, and further change the net emission factors. However, considering that those assumptions would likely add additional uncertainty and we do not discuss the uncertainties in emission factors, such indirect impacts on emission factors are not included in this study. We only

applied all the fuel consumption differences to the combustion sectors. The sectoral categories are consistent across all the energy datasets from NBS (Table S1). The same scale factor in fuel consumption was applied for all the sub-categories in same major sector (e.g. industrial coal-fired boilers and kilns in industry sector; on-road diesel vehicles and off-road mobile sources in transportation sector). The possible uncertainties in feedstocks and products resulted from energy uncertainties are not included in this study, and also the uncertainties in biomass consumption are not included due to lack of multiple datasets, thus our estimates of emission uncertainties are likely on the conservative side. As the emission calculations were performed with province-level data, energy consumption in the national energy statistics were directly allocated to provinces by using the ratios derived from the provincial energy statistics.

*The extent to which the results are interesting is derived from the scale of the inconsistencies of the underlying data, not from the analysis itself. While the authors appear positioned to undertake a more rigorous assessment of their important topic, the current paper is too formulaic, unsupported, and speculative to justify publication.*

**Response:** We agree that besides the "apparent uncertainty", the inconsistencies of the underlying data are also interesting, so we also presented and discussed the inconsistencies in the manuscript. For example, we presented the results from different datasets in Figure 1-3, Figure 5 and Table 3, and also discussed those inconsistencies in the results section and discussion section. In particular, we also discussed the inconsistencies of the energy data and possible sources for the inconsistencies in the section of "4.1 Understanding the reliability of energy statistics".

*Individual Questions/Issues*
*1. The paper is irreproducible, as it does not describe the methods of estimating emissions applied to different energy consumption datasets. It instead refers the reader to the website of the MEIC model, which does not present all of the underlying data and assumptions of the emission estimation model. To be reproducible, methods and assumptions must be described for each category of energy use (industrial subsector, for example, or vehicle type) treated uniquely in the assessment. Other researchers therefore cannot replicate the emission estimation as currently presented, except by blind trust in the same MEIC model.*

**Response:** We have clarified the question about reproducibility in the above response.

*2. The paper draws inferences about uncertainty based on two values defined in lines 9-10 of page 3: "We defined the apparent uncertainty as the maximum discrepancy among different datasets and the apparent uncertainty ratio as the ratio of the maximum discrepancy to the mean value from the different datasets." These two concepts sound attractive but the rationales for their use to draw inferences about statistical uncertainty are currently lacking in the paper.*

**Response:** We have clarified rationales for use of the metrics defined and employed in the paper in the introduction section of the revised manuscript. We defined the apparent uncertainty as the maximum discrepancy among different datasets and the apparent uncertainty ratio as the ratio of the maximum discrepancy to the mean value from the different datasets. Apparent uncertainty is a straightforward metric used to quantitatively gauge the apparent discrepancies between different existing datasets. Thus apparent uncertainty could partly reflect actual uncertainty. In general, large apparent uncertainty reflects large discrepancies, which might indicate large actual uncertainty. However, it should be noted

that apparent uncertainty could not fully represent actual uncertainty, and apparent uncertainty would likely to be conservative estimates as it might be subjected to the datasets used. Thus small apparent uncertainty does not necessarily mean to small actual uncertainty. Actual uncertainty, however, is difficult to be quantified and might need judgments.

*Both concepts appear problematic. Regarding "apparent uncertainty," a case has been made that a rough estimate of the uncertainty of energy data might be based on differences in values of subsequently revised data in the same official series (Marland et al. 2009). The rationale rests on a reasonable expectation that revisions represent increasing accuracy in the data and/or calculations, or "learning and convergence." In the current paper, however, any connection to this rationale is lost because the authors simply compile datasets from different series (national, provincial, and IEA) and seek a maximum differential. Some sort of conceptual rationale for readers to find meaning in the value defined as apparent uncertainty is required for this calculation to be interpretable.*

**Response:** It should be noted that this study focused on quantifying the discrepancy rather than giving best estimates. Although revisions might represent increasing accuracy, or "learning and convergence", actual uncertainty, however, is difficult to be quantified and might need judgments. An alternative approach is to use "apparent uncertainty" which is used in this study to present the apparent discrepancies observed from the existing datasets rather than estimating uncertainties based on judgments. Apparent uncertainty is a straightforward metric used to quantitatively gauge the apparent discrepancies between different existing datasets. This kind of apparent uncertainty exists not only in the energy datasets, but could also transfer to emission inventories. We believe "apparent uncertainty" is useful for understanding the "actual uncertainty". Apparent uncertainty could partly reflect actual uncertainty. In general, large apparent uncertainty reflects large discrepancies, which might indicate large actual uncertainty. However, it should be noted that apparent uncertainty could not fully represent actual uncertainty, and apparent uncertainty would likely to be conservative estimates as it might be subjected to the datasets used. Thus small apparent uncertainty does not necessarily mean to small actual uncertainty. We have clarified this in the introduction section of the revised manuscript.

*The "apparent uncertainty ratio" is problematic first because the numerator is apparent uncertainty, with the conceptual concern just noted, but then compounded by a denominator that is also hard to rationalize because of autocorrelation. Taking the mean of all datasets, including sequential revisions of the same dataset (CT-CESY-Ori, CT-CESY-1C, CT-CESY-2C, and CT-CESY-3C), implicitly assumes that they are independent. Without a defensible justification of this assumption, the calculations should recognize that revisions represent improving accuracy and should not be treated equally (as in a mean) in assessment of uncertainty.*

**Response:** The question about apparent uncertainty is clarified in above response. Apparent uncertainty ratio is a metric to quantify the relative deviation. As we did not perform best estimates, the mean of all datasets gives an alternative median estimate. We notice that changing the denominator from the mean to the newest revised dataset (i.e., CT-CESY-3C) do not significantly impact the calculations of the apparent uncertainty ratio.

*The paper requires a more rigorously conceived statistical basis to draw the sort of inferences about uncertainty that it seeks as its primary conclusions.*

**Response:** We have clarified that our statistical approach could provide indirect but still useful

information about uncertainty in the above responses.

*Technical Corrections:*
*The authors need to revisit the above fundamental issues first before they (and reviewers) put time into*
*other issues and technical corrections that this paper needs.*
**Response:** We have clarified all the issues in the above responses.

**Reference**

Akimoto, H., Ohara, T., Kurokawa, J., and Horii, N.: Verification of energy consumption in China during 1996-2003 by using satellite observational data, Atmos. Environ., 40, 7663–7667, doi:10.1016/j.atmosenv.2006.07.052, 2006.

Chen, Y., Zhang, Y., Fan, J., Leung, L. R., Zhang, Q., and He, K.: Application of an Online-Coupled Regional Climate Model, WRF-CAM5, over East Asia for Examination of Ice Nucleation Schemes: Part I. Comprehensive Model Evaluation and Trend Analysis for 2006 and 2011, Climate, 3, 627–667, doi:10.3390/cli3030627, 2015.

Geng, G., Zhang, Q., Martin, R. V., van Donkelaar, A., Huo, H., Che, H., Lin, J., and He, K.: Estimating long-term PM2.5 concentrations in China using satellite-based aerosol optical depth and a chemical transport model, Remote Sens. Environ., 166, 262–270, doi:10.1016/j.rse.2015.05.016, 2015.

Guan, D. B., Liu, Z., Geng, Y., Lindner, S., and Hubacek, K.: The gigatonne gap in China's carbon dioxide inventories, Nat. Clim. Change, 2, 672–675, doi:10.1038/Nclimate1560, 2012.

Hu, J., Chen, J., Ying, Q., and Zhang, H.: One-year simulation of ozone and particulate matter in China using WRF/CMAQ modeling system, Atmos. Chem. Phys., 16, 10333–10350, doi:10.5194/acp-16-10333-2016, 2016.

Huo, H., Zheng, B., Wang, M., Zhang, Q., and He, K.: Vehicular air pollutant emissions in China: evaluation of past control policies and future perspectives, Mitig. Adapt. Strat. Gl., 20, 719–733, doi:10.1007/s11027-014-9613-0, 2015.

Kurokawa, J., Ohara, T., Morikawa, T., Hanayama, S., Greet, J.-M., Fukui, T., Kawashima, K., and Akimoto, H.: Emissions of air pollutants and greenhouse gases over Asian regions during 2000–2008: Regional Emission inventory in ASia (REAS) version 2, Atmos. Chem. Phys. Discuss., 13, 10049–10123, doi:10.5194/acpd-13-10049-2013, 2013.

Lei, Y., Zhang, Q., Nielsen, C., and He, K.: An inventory of primary air pollutants and CO2 emissions from cement production in China, 1990-2020, Atmos. Environ., 45, 147–154, doi:10.1016/j.atmosenv.2010.09.034, 2011.

Li, M., Zhang, Q., Streets, D. G., He, K. B., Cheng, Y. F., Emmons, L. K., Huo, H., Kang, S. C., Lu, Z., Shao, M., Su, H., Yu, X., and Zhang, Y.: Mapping Asian anthropogenic emissions of non-methane volatile organic compounds to multiple chemical mechanisms, Atmos. Chem. Phys., 14, 5617–5638, doi:10.5194/acp-14-5617-2014, 2014.

Li, X., Zhang, Q., Zhang, Y., Zheng, B., Wang, K., Chen, Y., Wallington, T. J., Han, W., Shen, W., Zhang, X., and He, K.: Source contributions of urban PM2.5 in the Beijing-Tianjin-Hebei region: Changes between 2006 and 2013 and relative impacts of emissions and meteorology, Atmos. Environ., 123, 229–239, doi:10.1016/j.atmosenv.2015.10.048, 2015.

Liu, F., Zhang, Q., Tong, D., Zheng, B., Li, M., Huo, H., and He, K. B.: High-resolution inventory of technologies, activities, and emissions of coal-fired power plants in China from 1990 to 2010, Atmos. Chem. Phys., 15, 13299–13317, doi:10.5194/acp-15-13299-2015, 2015.

Liu, J., Mauzerall, D. L., Chen, Q., Zhang, Q., Song, Y., Peng, W., Klimont, Z., Qiu, X., Zhang, S., Hu, M., Lin, W., Smith, K. R., and Zhu, T.: Air pollutant emissions from Chinese households: A major and underappreciated ambient pollution source, P. Natl. Acad. Sci. Usa., 113, 7756–7761, doi:10.1073/pnas.1604537113, 2016.

Lu, Z., Streets, D. G., Zhang, Q., Wang, S., Carmichael, G. R., Cheng, Y. F., Wei, C., Chin, M., Diehl, T., and Tan, Q.: Sulfur dioxide emissions in China and sulfur trends in East Asia since 2000, Atmos. Chem. Phys., 10, 6311–6331, doi:10.5194/acp-10-6311-2010, 2010.

Lu, Z., Zhang, Q., and Streets, D. G.: Sulfur dioxide and primary carbonaceous aerosol emissions in China and India, 1996-2010, Atmos. Chem. Phys., 11, 9839–9864, doi:10.5194/acp-11-9839-2011, 2011.

Marland, G., Hamal, K., and Jonas, M.: How uncertain are estimates of $CO_2$ emissions?, J. Ind. Ecol., 13, 4–7, 2009.

Sinton, J. E.: Accuracy and reliability of China's energy statistics, China Economic Review, 12, 373–383, 2001.

Streets, D. G., Bond, T. C., Carmichael, G. R., Fernandes, S. D., Fu, Q., He, D., Klimont, Z., Nelson, S. M., Tsai, N. Y., Wang, M. Q., Woo, J. H., and Yarber, K. F.: An inventory of gaseous and primary aerosol emissions in Asia in the year 2000, J. Geophys. Res., 108, doi:10.1029/2002jd003093, 2003.

Zhang, Q., Streets, D. G., He, K., Wang, Y., Richter, A., Burrows, J. P., Uno, I., Jang, C. J., Chen, D., Yao, Z., and Lei, Y.: NOx emission trends for China, 1995-2004: The view from the ground and the view from space, J. Geophys. Res., 112, D22306, doi:10.1029/2007jd008684, 2007.

Zhang, Q., Streets, D. G., Carmichael, G. R., He, K. B., Huo, H., Kannari, A., Klimont, Z., Park, I. S., Reddy, S., Fu, J. S., Chen, D., Duan, L., Lei, Y., Wang, L. T., and Yao, Z. L.: Asian emissions in 2006 for the NASA INTEX-B mission, Atmos. Chem. Phys., 9, 5131–5153, 2009.

Zhao, Y., Nielsen, C. P., Lei, Y., McElroy, M. B., and Hao, J.: Quantifying the uncertainties of a bottom-up emission inventory of anthropogenic atmospheric pollutants in China, Atmos Chem Phys, 11, 2295–2308, doi:10.5194/acp-11-2295-2011, 2011.

Zheng, B., Huo, H., Zhang, Q., Yao, Z. L., Wang, X. T., Yang, X. F., Liu, H., and He, K. B.: High-resolution mapping of vehicle emissions in China in 2008, Atmos. Chem. Phys., 14, 9787–9805, doi:10.5194/acp-14-9787-2014, 2014.

Zhang, B., Wang, Y., and Hao, J.: Simulating aerosol-radiation-cloud feedbacks on meteorology and air quality over eastern China under severe haze conditions in winter, Atmos. Chem. Phys., 15, 2387–2404, doi:10.5194/acp-15-2387-2015, 2015.

Zheng, B., Zhang, Q., Zhang, Y., He, K. B., Wang, K., Zheng, G. J., Duan, F. K., Ma, Y. L., and Kimoto, T.: Heterogeneous chemistry: a mechanism missing in current models to explain secondary inorganic aerosol formation during the January 2013 haze episode in North China, Atmos. Chem. Phys., 15, 2031–2049, doi:10.5194/acp-15-2031-2015, 2015.

---

## Author Response (AR2)

*Page 3, line 3*
*This paper actually does not "present a full evaluation of the uncertainties in China's energy statistics and their effects on emission estimates". It only presents a partial analysis, as it only evaluates partial uncertainty, as noted by the authors, and does not attempt a full uncertainty analysis of all possible uncertainties in the energy data. This should be clarified.*
**Response:** Thanks. We have replaced "present a full evaluation" by "present an evaluation" in the revised manuscript.

*Page 4*
*"MEIC emission inventory model (available at http://www.meicmodel.org) "*
*As far as the reviewer recalls, the MEIC model is not available at this web site, only inventory data is available, not the actual model (the web site is not loading at the moment). This should be clarified.*

[revised manuscript text omitted]